# Implementation of Computer-Vision-Based Farrowing Prediction in Pens with Temporary Sow Confinement

**DOI:** 10.3390/vetsci10020109

**Published:** 2023-02-02

**Authors:** Maciej Oczak, Kristina Maschat, Johannes Baumgartner

**Affiliations:** 1Precision Livestock Farming Hub, The University of Veterinary Medicine Vienna (Vetmeduni Vienna), 1210 Vienna, Austria; 2Institute of Animal Welfare Science, The University of Veterinary Medicine Vienna (Vetmeduni Vienna), 1210 Vienna, Austria

**Keywords:** Precision Livestock Farming, computer vision, farrowing prediction, pigs, behavior, animal welfare

## Abstract

**Simple Summary:**

The automated detection of the beginning and ending of nest-building behavior in sows might support the implementation of new management strategies in farrowing pens, i.e., temporary sow confinement in crates. This might improve sows’ welfare and reduce piglet mortality by limiting sow confinement in crates to only a critical period of piglets’ lives. The objective of this study was to predict farrowing with computer vision techniques to optimize the timing of sow confinement. In this study, we developed a computer-vision-based method for the automated detection of the beginning and ending of nest-building behavior in sows. The study included 71 Austrian Large White and Landrace × Large White crossbred sows and four types of farrowing pens. The beginning of nest-building behavior was detected with a median of 12 h 51 min and ending with a median of 2 h 38 min before the beginning of farrowing. It was possible to predict farrowing for 29 out of 44 animals. The developed method could be applied to warn the farmer when nest-building behavior starts and then to confine the sow in a crate when the end of nest-building behavior is detected. This could reduce labor costs otherwise required for the regular control of sows in farrowing compartments.

**Abstract:**

The adoption of temporary sow confinement could improve animal welfare during farrowing for both the sow and the piglets. An important challenge related to the implementation of temporary sow confinement is the optimal timing of confinement in crates, considering sow welfare and piglet survival. The objective of this study was to predict farrowing with computer vision techniques to optimize the timing of sow confinement. In total, 71 Austrian Large White and Landrace × Large White crossbred sows and four types of farrowing pens were included in the observational study. We applied computer vision model You Only Look Once X to detect sow locations, the calculated activity level of sows based on detected locations and detected changes in sow activity trends with Kalman filtering and the fixed interval smoothing algorithm. The results indicated the beginning of nest-building behavior with a median of 12 h 51 min and ending with a median of 2 h 38 min before the beginning of farrowing with the YOLOX-large object detection model. It was possible to predict farrowing for 29 out of 44 sows. The developed method could reduce labor costs otherwise required for the regular control of sows in farrowing compartments.

## 1. Introduction

In the societies of European Union (EU) member states, there is a growing concern over the welfare of farm animals. Specifically, phasing out and, finally, the prohibition of the use of cage systems in the EU was proposed in the Citizens’ Initiative “End the Cage Age”, which was signed by over 1.4 million EU citizens and supported by the European Commission [1]. Moreover, in recent years, national legislation on animal welfare in two EU member states imposed limitations on the use of crates in farrowing pens, i.e., to only the critical period of piglet lives in Austria [2] and to a maximum of 5 days in Germany [3]. These limitations will become mandatory for pig farmers in both countries in 2033 and 2036, respectively. As a consequence, pig farmers in the EU will have to adopt new management methods in farrowing pens, i.e., temporary sow confinement.

An important challenge related to the implementation of such a management system is the optimal timing of sow confinement in crates, considering sow welfare and piglet survival. Compared to free farrowing, temporary confinement appears to be beneficial for reducing piglet mortality, but confinement before farrowing prevents nest-building behavior. One of the possibilities is to confine sows in crates after the nest-building behavior has finished but before farrowing starts [4]. However, in practical farm conditions, due to the biological variability in gestation length [5], such precision in confining sows is only possible if farm staff perform time-consuming observations of a sow’s behavior, including in the night time.

Precision Livestock Farming (PLF) has the potential to automatize the monitoring of sow behavior in farrowing pens and to indicate the right timing of sow confinement in crates, i.e., at the end of nest-building behavior [6]. To date, three PLF technologies have been tested to predict the beginning of farrowing, i.e., infrared photocells, force sensors, and accelerometers [7,8,9,10]. It is possible to predict the beginning of farrowing, as reported in the research cited above, including in pens, with a possibility of temporary sow confinement [6], based on the increased level of activity of sows during nest-building. The farrowing prediction results based on an increased activity level, presented in Oczak et al. [6], indicated that for 70% of sows, an alarm was generated within a period of 48 h before the beginning of farrowing, while 61% of these alarms were generated between 6 and 13 h before the beginning of farrowing [6]. A promising next step to improve the performance of the farrowing prediction is a more detailed, automated analysis of behaviors that constitute nest-building behavior, e.g., pawing, rooting [7], or pain behavior, e.g., back arch, tail flick.

Cameras and computer vision have not been applied so far for the automated prediction of farrowing. This technology potentially has multiple advantages over the other sensors that have been used so far for farrowing prediction, i.e., it can be used to monitor animal behavior in a non-invasive way, the whole body of the animal can be monitored, multiple pens can be monitored with one camera sensor, and complex animal behavior can be automatically detected e.g., aggressive behavior [11,12]. Moreover, in our previous research, the calculation of Euclidean distance on the centroids of rectangles, indicating the location of detected sows with the object detection model RetinaNet, was proven as a reliable method for sow activity monitoring in farrowing pens [13]. Our method achieved good similarity (R^2^ > 0.7) to activity measurement based on an ear-tag accelerometer and a gold standard method, i.e., human labeling. Thus, as a first step in the development of computer vision methods for farrowing prediction, we aim to test if a similar performance of farrowing prediction—as with other sensor technology, i.e., ear-tag accelerometer [6]—can be achieved with computer vision applied for monitoring the activity level of sows.

In this study, we aim to select the optimal method of the You Only Look Once X (YOLOX) object detection algorithm, i.e., nano, tiny, small, medium, large, or extra large, for sow activity monitoring in real-time. In 2022, YOLOX is a state-of-the-art object detector, designed for real-time applications which surpass YOLOv3 [14] in terms of performance as one of the most widely used detectors in industry [15]. YOLOX also surpasses RetinaNet, used by us in previous research, in terms of speed and accuracy [16]. We hypothesize that YOLOX will provide an optimal trade-off between speed and accuracy for activity monitoring in farrowing pens. We aim to test the performance of YOLOX methods on unseen farrowing pens and animals. The second objective is to validate the previously tested Kalman filtering and fixed interval smoothing (KALMSMO) algorithm for farrowing prediction [6]. This aims to test if the KALMSMO algorithm can achieve a similar performance for farrowing prediction independently of whether the activity of sows was estimated based on an accelerometer data, as in our previous research [6], or image data.

## 2. Materials and Methods

### 2.1. Ethical Statement

Project PIGwatch was authorized by the Ethical Committee of the Austrian Federal Ministry of Science, Research and Economy and by the Ethical Committee of Vetmeduni Vienna (GZ: BMWFV-68.205/0082-WF/II/3b/2014) according to the Austrian Tierversuchsgesetz 2012, BGBl. I Nr. 114/2012.

### 2.2. Experimental Setup

#### 2.2.1. Animals and Housing

The observation was conducted in 2 stages at Medau, the pig research and teaching farm (VetFarm) of the University of Veterinary Medicine Vienna, Vienna, Austria. The first dataset was collected between June 2014 and May 2016, while the second one between December 2021 and July 2022. In total, 71 Austrian Large White sows and Landrace × Large White crossbred sows were included in the trials. The parity of animals included in the study ranged from 1 to 8, were between 1 and 4 years of age, and weighed from 198 to 384 kg. The sows were kept in four types of farrowing pens with the possibility of keeping a sow either unconfined or in a farrowing crate. Out of 71 sows, 9 were kept in SWAP (Sow Welfare and Piglet Protection) pens (Jyden Bur A/S, Vemb, Denmark), 9 in trapezoid pens (Schauer Agrotronic GmbH, Prambachkirchen, Austria), 9 in wing pens (Stewa Steinhuber GmbH, Sattledt, Austria), and 44 in BeFree pens (Schauer, Prambachkirchen Austria). None of the animals included in the observations were confined in a farrowing crate from the introduction to the farrowing pen until the end of farrowing.

The SWAP pens had an area of 6.0 m^2^. The pens had a solid concrete floor in the front (lying area) and a slatted cast iron floor in the back (defecation area). The pen had 2 troughs, one for the crated and one non-crated sow (Figure 1a). The trapezoid pens had an area of 5.5 m^2^. The pens had plastic flooring in the creep area and solid concrete flooring in the sow lying area in front of the trough (Figure 1b). The wing pens had an area of 5.5 m^2^. The pens were partly slatted with plastic elements and solid concrete elements (Figure 1c). The BeFree pens had an area of 6.0 m^2^. Similarly to wing pens, BeFree pens were partly slatted with plastic elements on the side of the pen and solid concrete elements in the center (Figure 1d). In all 4 pen types, a straw rack was mounted in the front area of the pen, in close proximity to the feed trough.

The sows were introduced to the farrowing pens approximately five days before the expected date of farrowing. The date was derived from the usual gestation length of sows (114 days), which could vary from 105 to 125 days [5]. Farrowing was not hormonally induced. The observational period was from the introduction of the sow to the farrowing room until the end of farrowing. The farrowing pens that were recorded as part of the first dataset, i.e., SWAP, trapezoid and wing, were located in the testing unit of the farm. The second dataset contained only BeFree pens which were located in the production unit of the farm. Both the testing unit and production unit of the VetFarm had an automatic ventilation system. The average temperature in the room was 22 °C. The sows were fed twice a day during the observational period. Water was provided permanently in the troughs via a nipple drinker or an automatic water-level system. To fulfill the need for adequate material to explore and for nest building, sows were offered straw in the aforementioned rack throughout their stay in the pens. The racks were half-filled in the morning and whenever the racks were empty.

#### 2.2.2. Video Recording

The behavior of sows was video-recorded from the introduction to farrowing pens until the end of farrowing with two-dimensional (2D) cameras in order to create a dataset that could be labeled. Each pen in the first dataset (SWAP, trapezoid, and wing pens) was equipped with one IP camera (GV-BX 1300-KV, Geovision, Taipei, China) locked in protective housing (HEB32K1, Videotec, Schio, Italy) hanging 3 m above the pen, giving an overhead view. In dataset 2, each IP camera (GV-BX2700, Geovision) was installed with a top view of 2 farrowing pens (BeFree). Additionally, above each farrowing pen in both datasets, infrared spotlights (IR-LED294S-90, Microlight, Moscow, Russia) were installed in order to allow for night recording. The images were recorded with a 1280 × 720 pixel resolution in MPEG-4 format, at 30 fps for dataset 1, while at 25 fps for dataset 2.

The cameras used for recording the first dataset (SWAP, trapezoid, and wing pens) were connected to a PC on which the Multicam Surveillance System (8.5.6.0, Geovision, Taipei, China) was installed. The system allowed for the simultaneous recording of images from 9 cameras. The PC had a processor Intel5, CPU 3330, 3 GHz (Intel, Santa Clara, CA, USA) with 4 GB of physical memory. The operating system was Microsoft Windows 7 Professional (Redmond, WA, USA). The first dataset was stored on exchangeable, external 2 and 3 TB hard drives. The cameras used for recording the second dataset (BeFree pens) were connected to a server for the storage of video data (Synology, Taipei, Taiwan) with 4 cores, 8 GB memory, and 260 TB storage.

### 2.3. Dataset

The dataset was composed of video recordings which were collected in a period from the introduction to the farrowing pen until 24 h after the end of farrowing. Dataset 1—which contained recordings of 27 sows in SWAP, trapezoid, and wing pens—was used for the training of YOLOX object detection models and contained 4667 h of video recordings. Dataset 2, which contained recordings of 44 sows in BeFree pens, was used for the testing of YOLOX object detection models, the calculation of activity of sows, and the implementation of KALMSMO farrowing prediction models. It contained 17,713 h of video recordings. The division of recorded videos on dataset 1 and 2 was motivated by the first objective of this study, i.e., to test the performance of YOLOX methods on unseen farrowing pens and animals. This dataset division allowed for the simulation of the expected performance of the models when implemented in a new environment. We could expect a similar division of datasets used for the practical implementation of PLF models, where the initial dataset is collected within a certain period of time, in a limited number of farm environments or pens, i.e., SWAP, trapezoid, and wing pens in our study. On this dataset, the models are trained for a particular purpose, such as the detection of sow locations. Then, the trained models are implemented in a new, previously unseen environment, i.e., dataset 2 in our study. In our study, the breed of sows and sow management, including feeding, were the same between both datasets. However, the type of farrowing pen and climate conditions differed between both datasets as they were collected in different sections, i.e., testing unit and the production unit of the farm.

### 2.4. Data Labeling

To create a reference dataset on the basis of which further data analysis could be performed, we labeled the time of the beginning of farrowing of each individual sow (n = 71). It was defined as the point in time when the body of the first born piglet dropped on the floor. The time of birth of the last piglet indicated the end of farrowing. Labeling software Interact (version 9 and 14, Mangold International GmbH, Arnstorf, Germany) was used to label the beginning and ending of farrowing in dataset 1. For labeling dataset 2, we used labeling software Boris (version 7.9.15) [17].

To label the images, which could be later used for training and validation of the object detection model, i.e., YOLOX, we selected the frames from dataset 1 and 2 which contained the most relevant information on sows. This general approach allows for the most efficient use of computational resources for the training of the object detection algorithm, but also reduces the heavy workload related to the manual labeling of objects in the images by human labelers [13].

#### 2.4.1. Dataset 1

Frame selection was performed according to the procedure described in Oczak et al. [13]. For the purpose of the selection of specific frames to be used for labeling, we applied the k-means algorithm described in Pereira et al. [18]. The k-means algorithm was used to select images with the least correlation. In dataset 1, three days were selected, i.e., the day of introduction to the farrowing pen, one day before the day of farrowing, and the day of farrowing with a total of 175,000,000 frames. Out of 175,000,000 frames, the k-means algorithm identified 14,242 frames that were the most different between each other (Table 1). We decided to select 14,242 frames as it was possible to label this number of images within 3.5 weeks by one labeler (40 h/week)—this was manageable with the resources available in this research project.

One object class was labeled by a trained human labeler on each frame out of the selected 14,242 frames, i.e., body of the sows (Figure 2). The Computer Vision Annotation Tool (version 3.17.0) was used to label the frames [19]. The sow’s body was labeled with a rectangle so that the center of an object was placed in the center of the rectangle.

#### 2.4.2. Dataset 2

The frame selection was performed similarly as for dataset 1 with the same k-means algorithm. However, we selected 500 frames from all the videos recorded for all sows in dataset 2—recorded within a period from introduction to the farrowing pen until one day after farrowing. Because in dataset 2 there were 2 sows under one camera view (Figure 1d), the number of frames used for the labeling of sows was increased to 1000 by masking the view on either the right or left of the BeFree pen (Figure 2d).

The election of a lower number of frames in dataset 2 (1000) than in dataset 1 (14,242) was motivated by the reduction in the manual workload needed for labeling the frames. The human labeling time was 1 day in dataset 2. As dataset 1 already contained 3 types of farrowing pens and a relatively large number of frames with animals, we assumed that for validating the models and re-training on a fourth environment from dataset 2, a lower number of images was needed.

In dataset 2, one object class was labeled on each frame out of the selected 1000 frames, i.e., body of the sows (Figure 2). The COCO annotator was used to label the frames (version 0.11.1) [20].

### 2.5. YOLOX

#### 2.5.1. The Model and the Methods

YOLOX is an updated version of the YOLO model. The updates include switching the detector to an anchor-free manner, the application of a decoupled head, and the leading label assignment strategy SimOTA. These updates allowed for state-of-the-art results across a large-scale range of models [15]. In our study, the model was used to automatically detect sow locations in farrowing pens.

The OpenMMLab toolbox was used to train, validate, and test the methods of YOLOX, i.e., nano, tiny, small, medium, large, and extra large (Table 2). OpenMMLab is an open source project which contains reimplementations of state-of-the-art computer vision algorithms for object detection, animal pose estimation, action recognition, and tracking. We decided on OpenMMLab, MMDetection (version 2.25.2) as this computer vision framework allows for the effective experimentation and comparison of implemented deep-learning architectures [21].

Parametrization of the YOLOX methods was used as implemented in MMDetection, i.e., optimizer stochastic gradient descent (SDG), with learning rate 0.01 and momentum 0.9. Similar images were augmented as implemented in MMDetection with mosaic, random affine, mixup, random horizontal flip and color jitter. No changes were made to the architecture of YOLOX methods, optimizer, or augmentations provided in MMDetection. Python version 3.8 was used with MMDetection.

#### 2.5.2. Experiments

We designed 2 experiments to test the performance of methods of the YOLOX algorithm (Table 2) in terms of generalization ability and inference speed. In both experiments, out of 15,242 labeled images, 9969 (65.4%) were randomly selected for the training set, 4273 (28%) for the validation set, and 1000 (6.6%) for the test set. In experiment 1, training and validation sets included images from dataset 1, while test set included images from dataset 2. Thus, in experiment 1, it was possible to test the generalization ability of the YOLOX on new unseen sows and farrowing pens (BeFree). In experiment 2, all 4 pen types and sows were represented in training, validation, and test sets (Figure 3).

Training was set to 100 epochs and was carried out on RTX Titan (NVIDIA, Santa Clara, CA, USA) on a server with 24 Core CPU AMD 3.2 GHz and 256 GB RAM with an evaluation of validation and test set performances on every 5 epochs. Performance of the models was evaluated with standard 12 COCO evaluation metrics, e.g., Average Precision (AP) and Average Recall (AR) [22]. The optimal model was selected by the highest value of the primary COCO challenge metric Average Precision AP on the test set in both experiments. The speed of inference for each YOLOX method was estimated by inferring sow locations with the MMDetection function inference_detector on 1000 frames in the test set.

### 2.6. Activity Level of Sows

The two best performing models, one from experiment 1 and a second one from experiment 2, were used to extract sow locations in the videos recorded during the sow observational period in BeFree pens. Bounding boxes indicating sow locations were extracted in 1 fps (60 fpm) out of the videos recorded in 25 fps; later, these data were down-sampled to 40 fpm, 20 fpm, 5 fpm, 1 fpm, 12 fph, and 4 fph. In the next step, the Euclidean distance was calculated between centroids of extracted bounding boxes on the range of frame rates from 1 fps to 4 fph, as described in Oczak et al. [13]. The range of frame rates was used to evaluate which is the minimum frame rate for the real-time application of a farrowing prediction algorithm so that there is no negative impact on the performance of the algorithm. The lower the frame rate, the less computational resources would be required in the practical application of the algorithm.

The Euclidean distance was further smoothed with a mean calculated on a sliding window of 24 h with 15 min steps, similar to the research of Oczak et al. [6] where the standard deviation was used on the same window size and the same step to process ear-tag accelerometer data (Figure 4). This allowed for the elimination of variation in activity related to diurnal rhythms.

### 2.7. Farrowing Prediction

To estimate the dynamics of activity of sows, the Kalman filtering and fixed interval smoothing (KALMSMO) algorithm was used, as described in Oczak et al. [6], with the same hyper-parameter values of the model. The KALMSMO algorithm was fitted to an input variable, i.e., Euclidean distance at a fixed interval of 48 h, and expanded recursively by 15 min steps until the trend in animal activity changed to significantly increasing. The increase in activity trend was indicated by Euclidean distance reaching a higher value than the upper confidence interval of the estimated trend (Figure 4c). Then, the “first-stage” alarm was raised. The preferred time frame for the “first-stage” alarms was within 48 h before the onset of farrowing, and the alarm was not supposed to be generated after the onset of farrowing [6]. The “second-stage” was raised when the trend in animal activity changed to significantly decreasing. This was indicated by the input variable reaching a lower value than the lower confidence interval of the estimated trend. This alarm could be interpreted as an indication that nest-building behavior had ended. The preferred time frame for the “second-stage” alarm was after the “first-stage” alarm (within 48 h before the onset of farrowing) and not later than the end of farrowing [6].

The impact of the frame rate of video data (1 fps to 4 fph) on the performance of farrowing prediction was estimated by an analysis of proportion of “first-stage” alarms generated within 48 h before the beginning of farrowing. The number of sows for which no “first-stage” alarms were raised was also considered. Similarly, the impact of frame rate on the proportion of “second-stage” alarms raised after the “first-stage” alarm and not later than the end of farrowing was analyzed in the context of the performance of farrowing prediction.

Analysis was performed with a commercial software package (MATLAB 2019b, The MathWorks, Inc., Natick, MA, USA) and function irwsm of CAPTAIN toolbox [23] was used to fit the KALMSMO algorithm.

## 3. Results

### 3.1. Selection of YOLOX Methods

Results of both experiment 1 and 2 revealed, as could be expected, that more complex models of YOLOX (YOLOX-small, YOLOX-medium, YOLOX-large, YOLOX-extra large) had better AP in both validation sets and test sets (Figure 5). Higher AP was achieved for these models after shorter duration of training than for simpler models (YOLOX-nano, YOLOX-tiny).

The performance of models in experiment 1 was generally worse than in experiment 2. In experiment 1, in the test set after 100 epochs of training, the AP of most methods of YOLOX was higher than 80, while in experiment 2, it was higher than 90. In Figure 5c, it is possible to observe that the AP of models on unseen BeFree pens started to decrease at approximately 70 epochs—this indicated that models started to overfit to the training set of dataset 1 and their generalization ability decreased after approximately the 70th epoch. In experiment 2, the model performance increased for 100 epochs, suggesting that training sets longer than 100 epochs might lead to a better sow detection performance. It is also clear that the addition of all environments of interest, in which it is desirable to infer the location of sows, to the training set, improves the performance of object detection models. This suggests that for the practical implementation of YOLOX for activity monitoring, it is better to include some images of sows in the training set from the environment where the algorithm will be implemented. However, to estimate the importance of the difference of 11.2 AP on the test set between the YOLOX model trained with BeFree pens to a model trained without these pens (Table 3), we further compared the impact of using both models to extract sow activity on the performance of farrowing prediction.

The best model for the detection of sows in farrowing pens in unseen environments, i.e., BeFree pens, was YOLOX-medium, which had the highest AP of 84.2 on the test set in comparison to the other models. In experiment 2, the model with the highest AP of 95.4 on the seen environment was YOLOX-large. Thus, for the extraction of centroids of sows, these two models were used. YOLOX-medium trained for 70 epochs and YOLOX-large trained for 100 epochs.

The speed of inference of YOLOX models ranged from 21 fps with YOLOX-extra large to 42 fps with YOLOX-nano on recorded videos. With a maximum frame rate of 1 fps used in this study, even with the most complex YOLOX-extra large model, it would be possible to monitor the whole farrowing production unit at VetFarm Medau, with one RTX Titan (20 pens) installed on a modern server.

### 3.2. Farrowing Prediction

The results of farrowing prediction with the KALMSMO models on the highest sampled frame rate of videos (1 fps) indicated the “first-stage” alarms with a median of 10 h 46 min and 12 h 51 min based on centroids of sows extracted with YOLOX-medium and YOLOX-large, respectively (Figure 6). “First-stage” alarms were raised slightly later (2 h 5 min) based on the application of the YOLOX-medium model trained in experiment 1. This was confirmed by a further analysis of distribution of alarms with the 1st quartile of “first-stage” alarms at 4 h 8 min in comparison to 6 h 2 min and the 3rd quartile at 16 h 17 min in comparison to 19 h 43 min.

Similarly, “second-stage” alarms were raised slightly later based on the application of the YOLOX-medium model trained in experiment 1 with a median very near to the beginning of farrowing at 2 h 17 min in comparison to 2 h 38 min based on the YOLOX-large trained in experiment 2. The other metrics of distribution of “second-stage” alarms represented a similar difference in timing between both models, i.e., first and third quartiles (Figure 6). These results showed that the difference in performance of 11.2 AP between both YOLOX models, first trained in experiment 1 and second in experiment 2, had little impact on the difference in timing of “first and second-stage” alarms.

However, the analysis also revealed that the “first-stage” alarms were not raised for 20 out of 44 (45%) sows when predictions of farrowing were based on centroids extracted with YOLOX-medium trained in experiment 1 (Figure 7). This result is much worse in comparison to only 13 out of 44 (30%) sows, when YOLOX-large trained in experiment 2 was used.

Analysis of the frame-rate impact on the performance of farrowing prediction showed that with the highest frame rate of 1 fps, we obtained the highest number of alarms for a model trained in experiment 1, with “first-stage” alarms raised for 29 animals out of 44 (66%) in the period of 48 h before the beginning of farrowing (Figure 7). With the reduction in the frame rate, the percentage of sows for which the “first and second-stage” alarms were generated in the desired time decreased to a minimum of 11 sows (25%). The results of farrowing prediction in relation to the frame rate revealed the highest number of true positive alarms for YOLOX-large trained in experiment 1 at 20 fpm, and decreased with the lower or higher frame rate. However, the lowest number of true positive alarms was achieved based on YOLOX-large detections when the frame rate decreased below 1 fpm (Figure 7).

The highest number of true positive alarms was achieved with a frame rate of at least 20 fpm. With this frame rate, either the best performance was reached (model trained in experiment 1) or the performance increased only slightly with a further increase in the frame rate (model trained in experiment 1). Generally, a lower number of true positive alarms with lower frame rates of videos was related to increased variability in the calculated activity of sows. With lower frame rates, an increase in activity related to nest-building behavior and approaching farrowing became less distinguishable from normal activity patterns. This was visualized in Figure 8, in which we presented the results of the calculation of normalized Euclidean distance.

For all tested frame rates and both object detection models, there was only one sow (2%) for which a “first-stage” false alarm was raised, i.e., earlier than 48 h before the beginning of farrowing.

## 4. Discussion

Making PLF algorithms work in real-life conditions is a hard job that all developers will experience before successfully implementing a reliable and accurate real-time monitoring system [24]. Before implementing the image-based farrowing prediction system, it was necessary to validate a state-of-the-art object detection algorithm, i.e., YOLOX [15], decide on the frame rate of input video data, and finally validate the previously used KALSMO algorithm on the activity of sows extracted from image data [6].

In our former research, we applied the RetinaNet object detection model to detect sows in three types of farrowing pens [13]. We decided on this model because in PLF applications, accuracy but also real-time processing is of high importance and RetinaNet matched the speed of previous one-stage detectors while surpassing the accuracy of all existing state-of-the-art two-stage detectors at the date of the algorithm’s publication in 2017 [16]. However, between 2017 and 2022, the field of computer vision and modeling for object detection made significant progress. This is apparent from the comparison of performance of RetinaNet and YOLOX on reference dataset MSCOCO val [22] on which RetinaNet achieved 40.8 AP, while YOLOX 51.2 AP. The advantage of YOLOX over RetinaNet was also confirmed by comparing the results of our current study with the previous study, i.e., Oczak et al. [13]. In our current study, the detection of sows with YOLOX-large resulted in 96.9 AP, while RetianNet had 37 AP on exactly the same validation set.

In the study of Küster et al. [25], YOLOv3 was used to detect different parts of sows’ bodies in farrowing pens, i.e., heads, tails, legs, and udder, but not the whole sows’ bodies, as in our study. They detected heads with 97 AP_50_, tails with 78 AP_50_, legs with 75 AP_50_ and udder with 66 AP_50_. The performance of YOLOX-large used in our study was better in detecting the whole bodies of sows, with a perfect result of 100 AP_50_.

YOLOv3 was also used in the study of van der Zande et al. [26]. However, in this study, the model was used to detect the whole bodies of piglets in a group pen. A near-perfect AP_50_ of 99.9 was achieved but the authors applied post-processing on the results of object detection, i.e., the removal of false positives and removal of detections with a probability lower than 0.5. It is not clear from the article of van der Zande et al. [26] whether this post-processing affected the results of object detection AP_50_ metric. Nevertheless, such post-processing was not applied in our study and YOLOX had a perfect performance in detecting sows with 100 AP_50_.

These two comparisons of performance of YOLOX with YOLOv3 confirmed that the updates to the YOLOv3 made by the authors of YOLOX, such as switching the detector to an anchor-free manner or application of a decoupled head, improved the performance of YOLOv3 not only on reference dataset MSCOCO test-dev (51.2 AP and 42.4 AP) but also on our dataset with sows in farrowing pens. Moreover, YOLOX was also faster than YOLOv3 on MSCOCO test-dev [22] by 12.3 fps (57.8 fps and 45.5 fps). In our study, the speed of YOLOX was from 21 fps with YOLOX-extra large to 42 fps with YOLOX-nano. A comparison to YOLOv3 in the research of Küster et al. [25] and van der Zande et al. [26] was not possible as the speed of inference of YOLOv3 was not reported by the authors.

In both studies, the first of Küster et al. [25] and the second of van der Zande et al. [26], the validation of object detection models was performed on the same pens as the training of the models. This revealed the performance of the trained models in these pens. However, in PLF, we aim to implement the trained models on multiple farms, in different pens and environments [27]. One of the strongest arguments against using deep learning models (RetinaNet, YOLOv3, and YOLOX), also in the context of PLF research, is that in such over-parameterized and non-convex models, the system is easy to get stuck into local minima that generalizes badly on new datasets, e.g., with new farms or pens [28]. To test YOLOX’s generalization ability, we validated the models not only on seen but also unseen farrowing pens, i.e., BeFree. The difference between the performance of both models on the test set with BeFree pens, the first trained on BeFree and the second trained on the other farrowing pens, was 11.2 AP, indicating that the model which was trained on BeFree pens had a better performance. However, such a difference in AP metrics between both models is difficult to intuitively interpret. Additionally, the main objective of our work was to develop a system of farrowing prediction, not sow detection, which could work on additional farms. Thus, it was important for us to test what impact such a low performance of the model for sow detection could have on the performance of farrowing prediction.

The analysis of this impact revealed that 11.2 lower AP resulted in 20 sows out of 44 (45%) without the “first-stage” alarm, which is 7 additional sows to the 13 sows that already did not get the “first-stage” alarm when a YOLOX model trained on BeFree pens was used. This means that we can expect approximately a 50% increase in the number of sows without the “first-stage” alarm if we do not train an object detection model with a dataset collected on additional farms. The percentage of sows without the “first-stage” alarm could possibly be even higher on unseen farms and in unseen farrowing pens as additional factors such as different body color of sows of the other breeds could negatively affect the performance of object detection models. To our knowledge, it is the first time that such an analysis has been carried out. In the other studies, the training and validation datasets had the same type of farrowing pens. In addition, computer vision had not been used; instead, the other sensor technologies were utilized [7,8,9,10].

The reason for the increased number of sows without the “first-stage” alarm was that the YOLOX model that was not trained on BeFree pens in some of the images falsely detected parts of the farrowing pens as sows. This increased the variability in estimated activity of sows as the Euclidean distance was calculated between correctly detected sow centroids and centroids of false detections in consecutive frames. A similar effect of increased variability in the estimated activity level of sows on the decrease in performance of farrowing prediction can be observed in our study with lower frame rates than 20 fpm (Figure 8).

A promising approach to improve the performance of object detection models and further farrowing predictions on additional farms and farrowing pens is to generate synthetic datasets with, e.g., farrowing pens or other breeds of sows and train the models on these synthetic datasets. In this way, we can potentially generate training examples in a controllable and customizable manner with various light conditions or camera perspectives and avoid the challenges of collecting and labeling data in the real world. A rendering engine requires computer time rather than human time to generate examples. It has perfect information about the scenes it renders, making it possible to bypass the time and cost of human labeling [29].

In our current study, we focused on the detection of the whole sow bodies for activity estimation. We decided to not focus on the detection of parts of sow bodies because in our previous study [13], detection of parts of sow bodies, i.e., nose, head and ears, provided a very similar estimate of activity level to activity level based on the detection of whole sow bodies. Additionally, it was the easiest to detect the whole bodies of sows, as indicated by the highest AP of detection of this object. However, detection of parts of sow bodies, e.g., nose, head, ears or legs, might be very helpful if the focus of the study is on detection of specific nest-building or pain behavior for farrowing prediction. An example might be the study of Oczak et al. [30], in which detection of the use of the hay racks by the sows before the beginning of farrowing was based on detection of the nose, head, and ears of the sows. The method achieved a 96% classification accuracy. Methods that are potentially more robust than object detection models to occlusions present in the farrowing pens are key body point detection models such as Residual Steps Network or Contextual Instance Decoupling [31,32]. These models can be used to estimate the location of different parts of sow bodies even if they are occluded as the models contain information on the spatial structure of key body points, e.g., sow’s nose in relation to sow’s ears. Facial expressions were used to estimate pain in horses [33]. The automated detection of sow faces might be useful for the estimation of pain levels in sows before or during farrowing as other pain behaviors such as back arching were shown to precede expulsion of piglets [34].

The performance of the farrowing prediction model KALMSO applied for the first time on ear-tag accelerometer data in Oczak et al. [6] was confirmed in our current study. In both studies, the parameters of KALMSMO, e.g., nose-to-variance ratio, CI limits, were the same. In our previous study, the “first-stage” alarms were raised in the 48 h period before the beginning of farrowing for 18 out of 26 sows (69%), while in this study, for 29 out of 44 sows (66%). The “second-stage” alarms were raised for 17 out of 26 sows (65%) within 48 h before the beginning of farrowing until the end of farrowing in results based on ear-tag accelerometer data, and 28 out of 44 (63%) in our current study. The results of both studies were also very similar in terms of timing of the “first and second-stage” alarms.

In summary, the KALMSMO algorithm seems to be robust when applied for the purpose of farrowing prediction. The performance of the model is similar when applied for estimation of changes in trends in sow activity independently of two sensor types, i.e., ear-tag accelerometer or a 2D camera. However, for a better performance, the object detection algorithm which is necessary for calculating the activity of sows must be trained on images recorded in the farrowing pens where the farrowing prediction system will be implemented.

The proposed practical application of the “two stage” alarms for farrowing prediction was described in detail in Oczak et al. [6] in the context of changes in animal welfare legislation in the EU [2]. Based on the “first-stage” alarms indicating the beginning of nest-building behavior, the farmer could prepare for approaching farrowing. Such automated monitoring should reduce the need for the laborious manual observation of farrowing compartments and provide more comfortable conditions for sows at a time when they are sensitive to outside disturbances. Additionally, nest-building material could be provided to the sow, e.g., in a hay rack, to stimulate the nest-building behavior [30]. When the “second-stage” alarm is generated, most of the nest-building activity of a sow should be finished. Thus, confining a sow in a crate after the “second-stage” alarm has been generated should create little risk for sows staying in crates during nest-building behavior, and especially during the peak of nest-building behavior.

According to the first objective defined for this study, we selected the optimal methods of YOLOX for the implementation of the farrowing prediction algorithm at VetFarm Medau and the other farms. In the next stage of our research, we intend to implement the YOLOX-large trained in experiment 2 of this study for the estimation of location of sows at VetFarm Medau. With 20 farrowing pens in the farrowing unit of the farm and 1 fps frame rate of videos, our server with 24 Core CPU AMD 3.2 GHz, 256 GB RAM, and 2× Nvidia RTX3090 24GB should be sufficient for real-time farrowing prediction. Data on the activity of sows and alarms on approaching farrowing will be stored in InfluxDB time series database (InfluxData Inc., San Francisco, CA, USA). A Grafana (Grafana Labs, New York, NY, USA) dashboard will be used for data visualization. InfluxDB with Grafana was previously used by Kupfer et al. [35] for the storage and visualization of PLF data.

To implement the farrowing prediction methodology on farms other than VetFarm Medau, we recommend the application of YOLOX-medium trained in experiment 2 of our study for 70 epochs. This model seemed to generalize better than the other models on new, unseen farrowing pens and it was trained on all four types of farrowing pens available in our dataset. The presence of all farrowing pens in the training set might improve the performance of the model in new, unseen environments.

## 5. Conclusions

The optimal YOLOX method for the implementation of the farrowing prediction algorithm at the research farm on which the dataset was collected in this study was the YOLOX-large. For implementation of the farrowing prediction methodology on farms other than the research farm, we recommend the application of YOLOX-medium trained in experiment 2 of our study for 70 epochs. The KALMSMO algorithm confirmed the results of farrowing prediction obtained in the previous study on ear-tag accelerometer data, considering that the object detection model had to be trained on the farrowing pens present in the validation dataset. The developed method could be applied to warn the farmer when nest-building behavior starts and then to confine the sow in a crate when the end of nest-building behavior is detected. This could reduce labor costs otherwise required for the regular control of sows in farrowing compartments.

## Figures and Tables

**Figure 1 vetsci-10-00109-f001:**
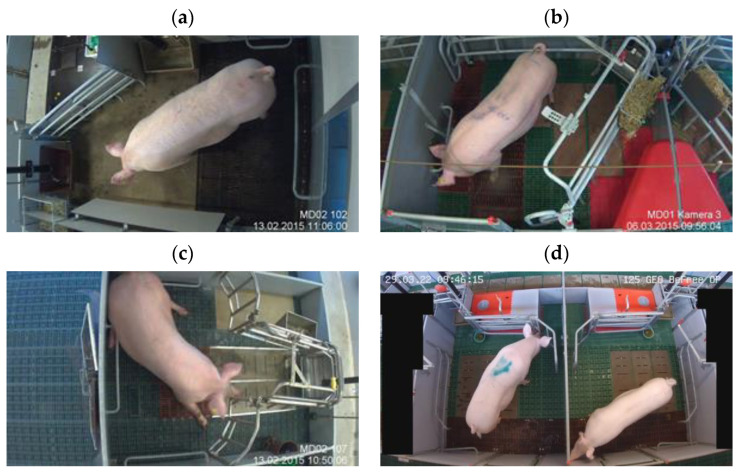
Farrowing pens with the possibility of temporary crating. (**a**) SWAP pen, (**b**) trapezoid pen, (**c**) wing pen, and (**d**) 2 BeFree pens.

**Figure 2 vetsci-10-00109-f002:**
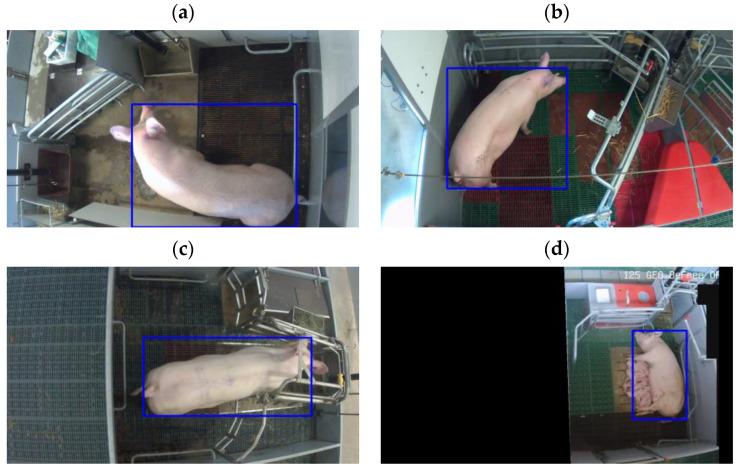
Labeled images with sows: (**a**) SWAP; (**b**) trapezoid; (**c**) wing; (**d**) BeFree (left pen under camera view is masked).

**Figure 3 vetsci-10-00109-f003:**
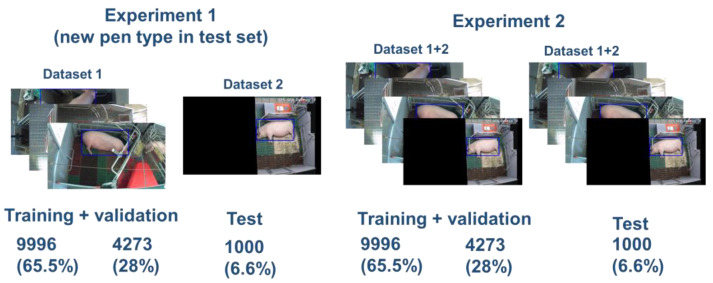
Data split in experiment 1 and 2. In experiment 1, it was possible to test the generalization ability of the YOLOX on new unseen sows and farrowing pens (BeFree). In experiment 2, all 4 pen types and sows were represented in training, validation, and test sets.

**Figure 4 vetsci-10-00109-f004:**
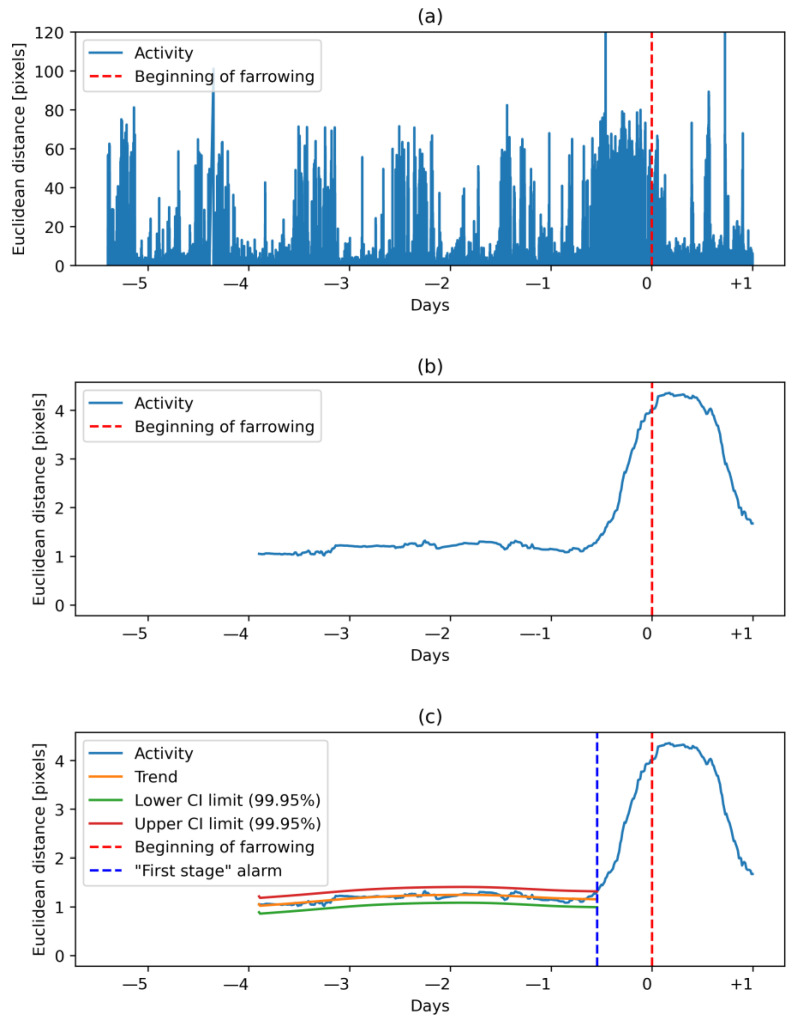
Euclidean distance calculated on centroids of detected sows. The period depicted in the plot starts at the introduction of a sow to a farrowing pen and ends around 1 day after the beginning of farrowing. (**a**) Euclidean distance calculated on consecutive frames. Peak activity was a few h before the onset of farrowing. (**b**) Mean of Euclidean distance calculated with a sliding window of 24 h and 15 min steps. The first 12 h of input variable was removed. Variation in activity related to diurnal rhythms was smoothed out. (**c**) Activity trend estimated on an expanding window of Euclidean distance. The “first-stage” alarm was at 13 h, before the beginning of farrowing.

**Figure 5 vetsci-10-00109-f005:**
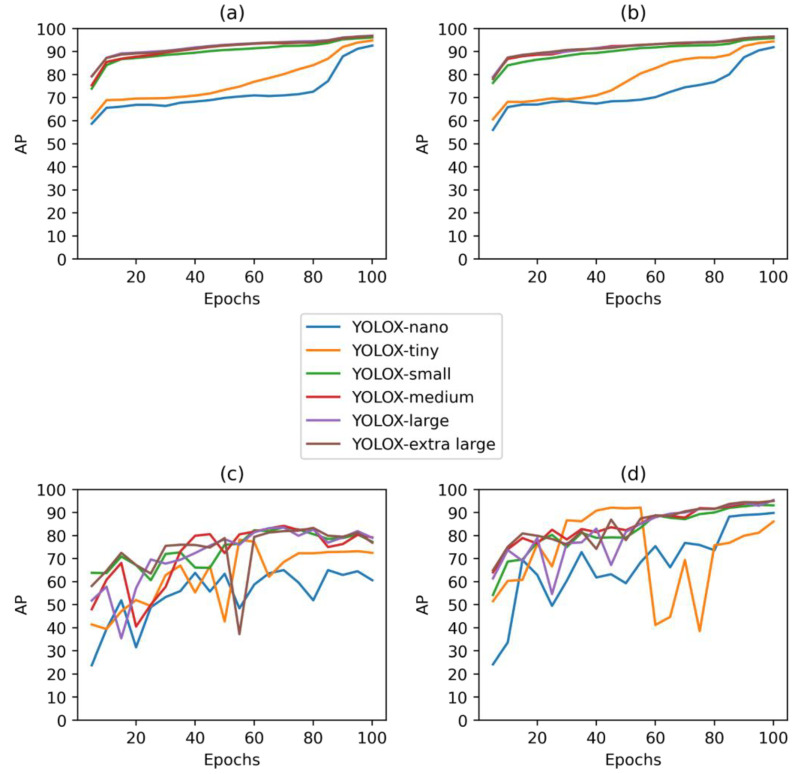
Performance metric AP on (**a**) validation set—experiment 1. (**b**) Validation set—experiment 2. (**c**) Test set—experiment 1. (**d**) Test set—experiment 2.

**Figure 6 vetsci-10-00109-f006:**
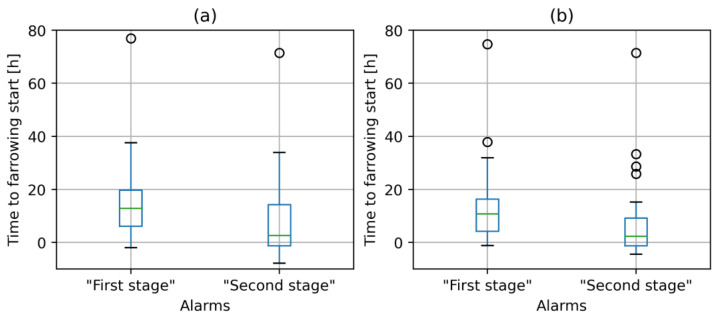
Distribution of duration between the time of alarms and beginning of farrowing. Circles indicate outliers. (**a**) Centroids of sows were extracted with YOLOX-medium trained in experiment 1. (**b**) Centroids of sows were extracted with YOLOX-large trained in experiment 2. The activity of sows was estimated on images sampled in 1 fps.

**Figure 7 vetsci-10-00109-f007:**
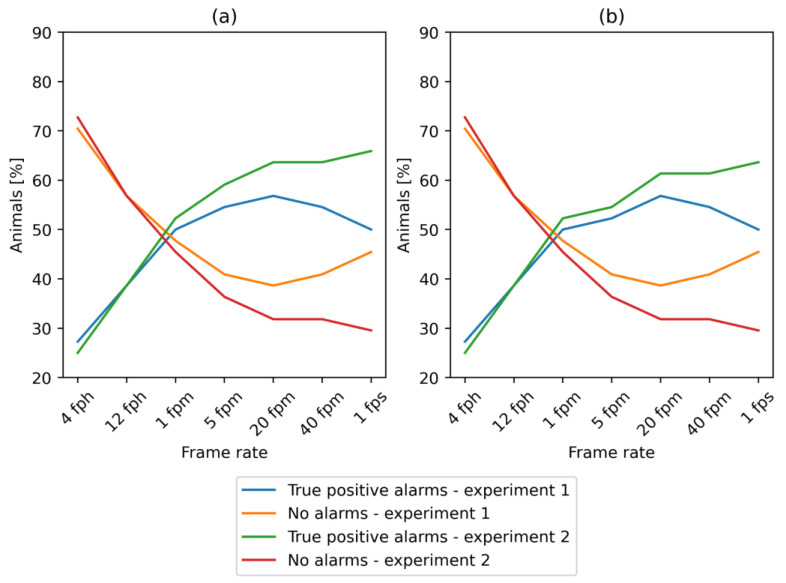
Results of farrowing prediction. Percentage of animals with (**a**) ”First-stage” alarms. (**b**) “Second-stage” alarms.

**Figure 8 vetsci-10-00109-f008:**
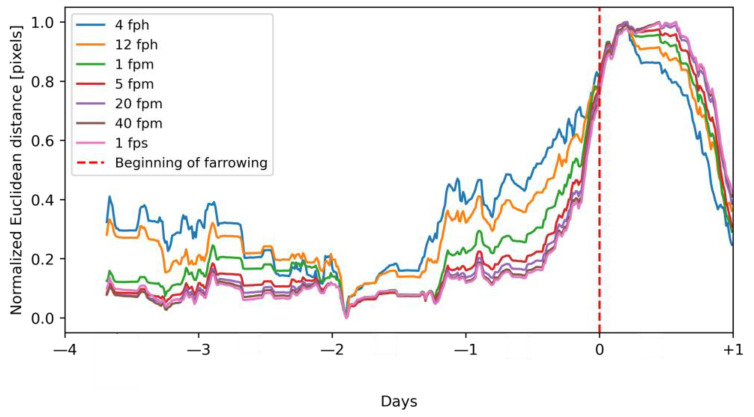
Normalized Euclidean distance of activity related to nest-building behavior.

**Table 1 vetsci-10-00109-t001:** Selected frames with the k-means algorithm for dataset 1 and 2.

Dataset	Duration of Video Recordings (h) ^1^	Frames Selected from Periods	N. Frames Selected
1	4667	Introduction to farrowing pen, one day before farrowing, day of farrowing.	14,242
2	17,713	From introduction to farrowing pen to one day after farrowing.	1000

^1^ Duration of video material recorded in the experimental period in dataset 1 and 2.

**Table 2 vetsci-10-00109-t002:** Methods of YOLOX.

Method	Parameters [millions]	AP on COCO ^1^
Nano	0.91	25.3
Tiny	5.06	32.8
Small	9.0	39.6
Medium	25.3	46.4
Large	54.2	50.0
Extra large	99.1	51.2

^1^ Mean Average Precision on benchmark dataset COCO [22].

**Table 3 vetsci-10-00109-t003:** Best performance of YOLOX methods evaluated by AP.

Experiment	Dataset	Method	Epoch	AP	AP_50_	AP_75_
1	Validation	YOLOX-large	100	96.9	99.0	98.9
1	Test	YOLOX-medium	70	84.2	99.0	98.9
2	Validation	YOLOX-medium	100	96.5	100	99.0
2	Test	YOLOX-large	100	95.4	99.0	98.9

## Data Availability

The data presented in this study are available on request from the corresponding author. The data are not publicly available due to the large size of collected video data, which was more than 3 TB.

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
