# Peer review of "Implementation of Computer-Vision-Based Farrowing Prediction in Pens with Temporary Sow Confinement"

_vetsci, 2023, doi:10.3390/vetsci10020109_

Round 1
Reviewer 1 Report
Currently, studies focusing on precision livestock farming are highly relevant for animal production and welfare since these technologies do not only can help to improve productive parameters but can also be implemented into specific events to assess animal welfare, such as the present study. I left some comments per section since there were no line numbers.
Abstract: In this section, it could be included that adopting new management methods could also improve animal welfare during farrowing for both the sow and the piglets.
It is important to add an appropriate objective of the present study before describing the methods and results, and a conclusion according to your results.
Also, please, add the breed of the sows
Keywords: Consider adding “animal welfare”
Introduction: In this section, when mentioning the welfare and survival of the piglets, some factors that can affect the farrowing process and survival of the neonates could be added to highlight the importance and application of the technology that the authors applied.
Additionally, the objective stated here needs to be added to the simple summary and abstract.
Material and methods: Some other characteristics of the animals could be added such as the age, parity, weight, and any other inclusion/exclusion criteria for the sows.
Results: On page 13, please, revise if the text “only slightly with further increase of the video frame rate” is correctly placed above Figure 8.
Discussion: Since you used whole sows’ bodies, it would be interesting to discuss the difference between evaluating only one part of the body with YOLO and the whole body. If there are significant differences or advantages/disadvantages between one or another. Likewise, it could be added if the detection of a part of the body, for example, the face, could be implemented in other research fields such as pain recognition and pain-related facial expression.
Reference list: Revise the journal’s instructions for authors and amend the list.
Reviewer 2 Report
Very interesting study in line with future animal welfare requirements for pigs.
Few notes:Change word " livestock" to "farm animals"
Above figure 8 there is a strange text "only slightly with further increase of the video frame rate"
Specific comments
Change word " livestock" to "farm animals", my opinion is that “farm animals” is more suitable
Figure 3 should be described more precisely.
Above figure 8 there is a strange text "only slightly with further increase of the video frame rate"
Title of Figure 8 should be changed to “Normalized Euclidean distance of activity related to nest-building behaviour.
There is no need to additionally explain details in text of Figure 8
Comments like “This reduces the performance of farrowing prediction“ should be included in Discussion
Some of technical data in Discussion, like those of server with 24 Core CPU AMD 3.2GHz, 256 GB RAM and 2xNvidia RTX3090 24GB should be also included in Material and Methods
In this specific passage inside Discussion the same reference is used six times ?! Try to change this and reduce citation of the same paper.
„Performance of the farrowing prediction model KALMSO applied for the first timeon ear-tag accelerometer data in Oczak et al. [6] was confirmed in our current study. Inboth studies the parameters of KALMSMO e.g. nose-to-variance ratio, CI limits, were thesame. In Oczak et al. [6] the “first-stage” alarms were raised in the 48 h period before thebeginning of farrowing for 18 out of 26 sows (69%), while in this study for 29 out of 44sows (66%). The “second-stage” alarms were raised for 17 out of 26 sows (65%) within 48h before the beginning of farrowing until the end of farrowing in results of Oczak et al. [6]and 28 out of 44 (63%) in our current study. The results of both studies were also verysimilar in terms of timing of the “first and second-stage” alarms. In Oczak et al. [6] the“first-stage” alarms were raised with a median of 8 h 51 min, while in the current studywith a median of 12 h 51 min. Difference of 4 h in the medians might be due to thefarrowing pens’ construction or individual variability between the animals as in Oczak etal. [6] difference in median of timing between the “first-stage” alarms in 3 types offarrowing pens (SWAP, trapezoid and wing) were maximum 2 h. In Oczak et al. [6] the“second-stage” alarms were raised with a median of 2 h 3 min, while in the current studywith a median of 2 h 38 min.“
